Evaluation of enamel integrity after clear aligner attachments removal: a risk–benefit comparison of one-step tools

Nguyen Anh Viet
Nguyen Trang Thi trang.nguyenthi@phenikaa-uni.edu.vn
Faculty of Dentistry, Phenikaa University , Hanoi , Vietnam
Gkantidis Nikolaos
Electronic publication date: 2025 Oct 15
Publication date: 2025
Volume: 13
Electronic Location ID: e20200
Received 2025 May 19; Accepted 2025 Sep 16
Copyright: ©2025 Nguyen and Nguyen
Copyright year: 2025
Copyright holder: Nguyen and Nguyen
License: This is an open access article distributed under the terms of the Creative Commons Attribution License, which permits unrestricted use, distribution, reproduction and adaptation in any medium and for any purpose provided that it is properly attributed. For attribution, the original author(s), title, publication source (PeerJ) and either DOI or URL of the article must be cited.
License URL: https://creativecommons.org/licenses/by/4.0/

Keywords: Dental enamel, Orthodontic appliances, Dental debonding, Dental instruments, Tooth surface

Funding: The authors received no funding for this work.

==============================
Background

At the end of clear aligner treatment, attachment removal is essential to restore natural tooth morphology while preserving enamel integrity. This study aimed to compare the effects of seven different composite removal instruments on the enamel surface using a risk–benefit model.

Materials

The instruments assessed were: OneGloss, Enhance, SM 104, Sof-Lex disc (coarse grit), tungsten carbide bur, zirconia bur, and white stone bur. Seventy extracted premolar teeth were bonded with standardized condensable composite attachments (2 × 4 × 1 mm). Enamel integrity was quantitatively assessed by measuring surface roughness (µm) and enamel loss depth (mm), and qualitatively analyzed using scanning electron microscopy (SEM). The time required for attachment removal was recorded in seconds. A risk–benefit model was applied to compare enamel damage and time efficiency across tools.

Results

White stone bur exhibited higher surface roughness and greater enamel loss compared to other groups (p = 0.000). Aluminium oxide-based tools (OneGloss, Enhance, SM 104, and the Sof-Lex disc) resulted in less enamel loss than the tungsten carbide bur. The risk–benefit model indicated that OneGloss, Enhance, and SM 104 offered the best balance between enamel preservation and time efficiency. SEM analysis further confirmed that OneGloss and SM 104 produced smoother enamel surfaces compared to other instruments.

Conclusion

OneGloss and SM 104 demonstrated superior time efficiency and preservation of enamel integrity compared to other evaluated tools.

Introduction

Clear aligners were introduced as an orthodontic treatment method to replace traditional brackets, offering improvements in both aesthetics and patient comfort (Ke, Zhu & Zhu, 2019). Similar to traditional fixed orthodontics, clear aligners also use auxiliary tools known as attachments, which are bonded to the tooth surface to aid in and control tooth movement (Morton et al., 2017). These attachments can be made from various types of composite materials. The lower viscosity of flowable nanocomposites allows for better adaptation during clinical attachment fabrication; however, it negatively affects their mechanical properties. In contrast, condensable composites demonstrate superior performance and resistance to mechanical stresses compared to flowable materials, making them more suitable for clinical applications (Chen et al., 2021; Gazzani et al., 2022).

Upon completion of orthodontic treatment, the removal of composite attachments and bracket bonding adhesives is a crucial clinical step. The primary objective is to restore the enamel surface as closely as possible to its pre-treatment condition, preserving long-term dental health and esthetics (Ryf et al., 2012). However, this process is not without challenges. One major concern is the risk of iatrogenic enamel damage, which can manifest as increased surface roughness, enamel loss, or the formation of microcracks. Residual attachment cement left on the tooth surface can also provide a site for bacterial plaque accumulation and staining over time (Fan, Chen & Huang, 2017; Ghaleb et al., 2024). Additionally, dental professionals face efficiency pressures, as chairside treatment time can impact both patient experience and clinic productivity (Ryf et al., 2012). Therefore, selecting appropriate tools and techniques for attachment removal is crucial to achieving optimal results, balancing effective removal with maximum enamel preservation.

Various tools and methods for removing residual cement from the tooth surface have been introduced, and their effectiveness has been evaluated in experimental models. While tungsten carbide burs are commonly used for polishing and removing excess composite after bracket removal due to their time efficiency, studies have shown that they can cause irreversible enamel damage, even when used with polishing instruments (Janiszewska-Olszowska et al., 2016; Ryf et al., 2012). Zirconia burs, another high-strength material designed for cement removal, have been shown to perform comparably to tungsten carbide burs in removing adhesive remnants (Thawaba et al., 2023). Additionally, many polishing tools have been developed using aluminum oxide as the abrasive material. Examples include the OneGloss system, which has a silicone base, and the Enhance system, which uses a urethane dimethacrylate base. These systems integrate both polishing and adhesive removal functions, operating through an altered pressure mechanism. Studies have shown that these one-step polishing systems effectively remove residual adhesive while restoring a smooth enamel surface (Fan, Chen & Huang, 2017; Shah et al., 2019; Thys et al., 2022). Another system that also utilizes aluminum oxide as an abrasive is the Sof-Lex polishing discs. These discs or strips incorporate abrasive particles, and their grit size determines the level of abrasiveness, with gradual changes in grit size affecting surface smoothness. Research indicates that using Sof-Lex as a multi-step polishing system yields less surface roughness than the tungsten carbide bur (Balachandran et al., 2016), but higher than the OneGloss polisher and the Enhance system (Shah et al., 2019). However, no studies have evaluated its effectiveness as a standalone adhesive removal method when used in a one-step polishing approach. For white stone bur, although they also rely on the abrasive action of aluminum oxide grains, these particles are tightly bonded in a stone-like structure, allowing for a consistent abrasive effect and precise performance. However, a study by Thawaba et al. (2023) found that while white stone burs efficiently remove residual adhesive in less time, they also cause significant enamel damage.

Studies evaluating the effectiveness of adhesive removal tools typically rely on two main criteria: tool efficiency, measured by the time required for complete cleaning, and enamel integrity, which can be assessed through surface roughness parameters such as Ra and Rz, using a profilometer. Furthermore, changes in enamel morphology can be qualitatively observed using scanning electron microscopy (SEM), which enables surface examination at magnifications of up to several thousand times (Chen et al., 2021; Fan, Chen & Huang, 2017; Gazzani et al., 2022; Ghaleb et al., 2024). In addition to surface roughness, gross enamel changes can be quantified by measuring enamel loss through pre- and post-polishing scan superimposition. This method provides an average enamel loss measurement and visual representations of enamel surface changes (Janiszewska-Olszowska et al., 2015; Ryf et al., 2012).

While numerous studies have explored adhesive removal methods after bracket debonding, the clinical scenario of removing composite attachments used in clear aligner therapy remains underexplored. Unlike bracket bonding, which often involves forceful debonding, clear aligner attachments use different composite types, and the removal process does not apply mechanical force directly to the enamel. These distinctions suggest that attachment removal may present different challenges and risks to enamel integrity, underscoring the need to evaluate tool performance specifically for this context.

Given the clinical importance of balancing efficiency with enamel preservation, a systematic approach to tool selection is essential. Therefore, this study aimed to evaluate the effects of seven one-step composite removal tools on enamel surface quality following clear aligner attachment debonding. A risk–benefit model was introduced to weigh time efficiency against enamel damage, offering a practical decision-making framework for clinicians.

Materials and Methods

Study design

Written informed consent was obtained from all patients whose teeth were used in this study. Ethical approval was granted by the Ethics Committee of Phenikaa University (Approval No. PU2024-3-D-02).

The total sample size was determined based on an effect size of 1.71, as referenced from a previous study (Thawaba et al., 2023), using G*Power software (Faul et al., 2007) with α error = 0.05 and power = 90.0%. This calculation resulted in a sample size of 10 teeth per group.

Sample preparation

The study was conducted on first premolars freshly extracted for orthodontic purposes from July 2024 to February 2025. The teeth were cleaned under running water using a brush to remove any surface debris and then disinfected by immersion in 0.1% thymol solution for 48 h. After disinfection, they were stored in distilled water to prevent dehydration until use. Each tooth was carefully examined under a dental light to ensure it met the following inclusion criteria: No cracks or fractures, no caries on the buccal surface, no enamel hypoplasia, no prior orthodontic treatment, no previous restorations, and sufficient surface area for attachment bonding.

Baseline surface roughness measurement

Before attachment bonding, the tooth surfaces were thoroughly cleaned and polished with a rubber cup and non-fluoridated pumice, rinsed with distilled water, and dried using oil-free compressed air. The baseline surface roughness was measured in µm using a contact profilometer (SRT-6200, Landtek, China), using the Ra parameter. The sensor tip was moved along the longitudinal axis of the tooth crown, and each measurement was repeated three times; the average was recorded as the baseline surface roughness. These baseline values served as a reference (control) for evaluating changes in surface roughness after treatment. Surface roughness was measured independently by two examiners, who were blinded to the group assignments to minimize measurement bias. The final value was calculated as the average of their measurements. The teeth were also scanned using an intraoral scanner (Medit i900, MBK Partners, Seoul, South Korea).

Attachment bonding

A vertical rectangular attachment (2 × 4  ×1 mm) was digitally designed using Meshmixer (Autodesk, San Francisco, CA, USA), positioned at the center of the tooth crown with its long axis aligned to that of the tooth. Custom attachment placement templates were fabricated using a 0.64 mm aligner sheet (GS020, Amedes, Hamburg, Germany) based on 3D-printed tooth models. The attachment bonding procedure followed this protocol: A 37% phosphoric acid etchant (Total Etch, Ivoclar, Schaan, Liechtenstein) was applied to the enamel surface for 30 s, followed by rinsing with water for 10 s and gentle air-drying. A thin layer of bonding agent (Transbond XT, 3M, St. Paul, MN, USA) was then applied for 20 s, air-dried, and light-cured for 10 s using an LED curing unit (Woodpecker LED.F). The attachment template was filled with a condensable composite resin (Escom 100, Spident, Incheon, South Korea) and positioned over the tooth. A calibrated force of 20N was applied with a universal testing machine (WDW-1, Better United, Hebei, China) to ensure proper adaptation, and the composite was light-cured for 30 s before the template was carefully removed. The teeth were stored in distilled water for 24 h before attachment removal.

After attachment bonding, each tooth was rescanned using the same method described above to record the attachment’s exact position for reference in enamel loss measurements.

Attachment removal process

After attachment bonding, the teeth were divided into seven groups, with composite removal performed using different polishing tools: (Group 1) OneGloss, (Group 2) Enhance, (Group 3) SM 104, (Group 4) Sof-Lex Extra-Thin Brown Coarse Disc, (Group 5) 12-flute egg-shaped Tungsten carbide bur, (Group 6) Zirconia multiblade bur, (Group 7) White stone bur (Fig. 1). Details about the composite removal tools are summarized in Table 1.

Figure 1 Attachment removing tools.

Table 1 Instruments for composite attachment removal.

Group	Name	Shape	Main components	Manufacture	
1	OneGloss	Inverted cone	Silicone-impregnated with aluminium oxide particles	Midi, Shofu, Kyoto, Japan	
2	Enhance	Disc	Polymerized urethane dimethacrylate resin with aluminium oxide particles	Dentsply, Milford, USA	
3	SM 104	Disc	Silicone-impregnated with aluminium oxide particles	Dian Fong,
Shenzhen, China	
4	Sof-Lex™ XT Discs	Disc	Aluminium oxide-coated paper/plastic discs	3M ESPE, Seefeld, Germany	
5	Tungsten carbide bur	Egg shape with 12 flutes	Tungsten carbide alloy	H379AGK, Komet, Germany	
6	Zirconia bur	Cylindrical shape with multiblade	Zirconium oxide ceramic	Morelli, Sorocaba, SP, Brazil	
7	White stone bur	Flame shape	Sintered aluminium oxide	Frank dental, Gmund, Germany	

Attachment removal was carried out using a low-speed handpiece set to 20.000 rpm for Groups 1-6, and a high-speed handpiece operating at 120.000 rpm for Group 7. To ensure consistency, the procedure was performed by the same operator throughout. Polishing continued until no adhesive remnants were visible macroscopically.

The duration required for complete composite resin removal was recorded in seconds. Following the removal process, the teeth were scanned again, adhering to the same protocol described earlier.

Post-operative surface roughness measurement

After complete composite resin removal, surface roughness was remeasured using the same protocol as at baseline. Each tooth was also rescanned following the previously described method.

Scanning electron microscopy analysis

Surface morphology was examined using a scanning electron microscope (Quanta 450, Thermo Fisher Scientific, Waltham, MA, USA). Two randomly selected samples from each group were examined. In addition, untreated enamel samples were included as a control reference to facilitate qualitative comparison. The samples were coated with conductive carbon using thermal evaporation (Leica EM SCD050, Leica Microsystems, Wezlar, Germany) before observation at ×150 and ×2,400 magnification.

Enamel loss measurement

Depth of enamel loss was quantified by superimposing three digital scans of each tooth: baseline scan (before bonding), post-bonding scan (after attachment placement), and post-removal scan (after attachment removal). The post-bonding scan was used to identify the attachment region and ensure measurement consistency across all samples. The scans were saved in STL format and imported into Medit Link software (Medit, Seoul, South Korea). The Best-Fit Alignment Tool was used to accurately align the three scans. The Medit Compare Tool analyzed the difference between the pre- and post-removal scans. A color-coded deviation map was generated to visualize surface changes. Mean surface distance was used to quantify the depth of enamel loss at specific areas in millimeters (Fig. 2).

Figure 2 Enamel loss measurement procedure.

Scans of the tooth before bonding, after bonding, and after attachment removal were imported into Medit Link software. These scans were superimposed to evaluate enamel loss in the area where the attachment had been placed.

Data analysis

All data were analyzed using SPSS software (version 23.0; IBM, Armonk, NY, USA). The Shapiro–Wilk test was employed to assess the normality of the data distribution. Results were presented as mean ± SD. Preoperative and postoperative surface roughness were compared using a paired-sample t-test. For comparisons of mean values across groups, one-way ANOVA was performed, followed by post hoc analysis using the Tukey test for multiple group comparisons. Statistical significance was determined at a threshold of p = 0.05. The graphs were generated using Python 3.12.8 version (https://www.python.org).

To assess intra-operator reliability, a random subset of 10 samples per investigator was re-evaluated after a one-week interval without access to the original measurements. Inter-operator reliability was determined by comparing measurements of surface roughness and enamel loss independently performed by both investigators on the same samples. Intraclass correlation coefficients (ICC) were calculated to evaluate measurement consistency.

A three-dimensional risk–benefit matrix was constructed to visually and comparatively assess the clinical performance of each instrument. The matrix incorporated three variables: two risk factors were enamel loss and surface roughness, and one benefit factor was time efficiency. Each criterion was scored on a scale from one to five based on the results of the Tukey test. A score of five represents the highest enamel loss, highest surface roughness, or fastest operating time; a score of three indicates intermediate results; and a score of one corresponds to the lowest enamel loss, lowest surface roughness, or slowest operating time.

Results

Intra-rater reliability

Operator calibration was performed to assess the test–retest reliability of measurements using a one-way random-effects model. For enamel loss, the intraclass correlation coefficient (ICC) for operator 1 was 0.95 (95% CI [0.79–0.99]), and for operator 2, it was 0.86 (95% CI [0.43–0.97]). For surface roughness (Ra), the ICCs were 0.93 (95% CI [0.72–0.98]) for operator 1 and 0.96 (95% CI [0.85–0.99]) for operator 2. These results indicate excellent intra-operator reliability and a high level of measurement consistency across repeated assessments.

Surface roughness measurement

The intraclass correlation coefficient (ICC) for Ra values before and after resin removal was approximately 1, indicating excellent inter-rater reliability. Table 2 summarizes the surface roughness measurement by Ra values (µm) for each tool at T0 (baseline) and T1 (post-adhesive removal).

Table 2 Enamel surface roughness (μm).

T0, pre-attachment bonding; T1, post-attachment removal. Surface roughness between T0 and T1 was compared using a paired-sample t-test. Surface roughness across different groups was analyzed using a one-way ANOVA test. According to Turkey tests, means with the same superscript letters were not significantly different.

Group	Polisher	Sample size	Surface roughness (µm)
(Mean ± SD)	p-value (paired
sample t-test)	
			T0	T1		
1	OneGloss	10	0.58 ± 0.16	0.54 ± 0.13a	0.51	
2	Enhance	10	0.73 ± 0.18	0.54 ± 0.15a	0.03	
3	SM 104	10	0.57 ± 0.18	0.56 ± 0.15a	0.91	
4	Sof-lex disc	10	0.57 ± 0.13	0.51 ± 0.11a	0.27	
5	Tungsten carbide bur	10	0.47 ± 0.06	0.57 ± 0.10a	0.01	
6	Zirconia bur	10	0.64 ± 0.19	0.61 ± 0.14a	0.71	
7	White stone bur	10	0.58 ± 0.19	0.81 ± 0.17b	0.001	
p-value (one-way ANOVA)		P = 0.183	p = 0.000		

One-way ANOVA and Tukey’s post hoc analysis showed no significant difference in surface roughness among groups before attachment removal (p = 0.065, partial eta squared (η2) = 0.167). However, there was a significant difference in surface roughness after adhesive removal (p = 0.000, η2 = 0.345). After composite removal, there was no significant difference among Groups 1–6. In contrast, all groups exhibited significantly lower roughness than Group 7 (p = 0.001).

Scanning electron microscopy analysis

Scanning electron microscopy (SEM) microphotographs revealed that the intact enamel surface, before attachment bonding, appeared generally smooth with occasional craters but no visible grooves or scratches. Following attachment removal, the enamel surface in Group 1 closely resembled that of Group 3, exhibiting a smooth texture with parallel, shallow grooves. Group 2 showed a surface characterized by relatively deep scratches and grooves, though without prominent craters. Group 4 produced a surface almost as smooth as those in Groups 1 and 3, though fine scratches and multidirectional grooves were evident. Group 6 created a relatively smooth surface with shallow grooves and small craters. Group 5 and Group 7 resulted in a rougher surface, characterized by shallow, irregular grooves in varying directions and larger craters (Fig. 3).

Figure 3 Scanning electron image (SEM) analysis.

SEM images of enamel surfaces before (A-1, A-2) and after attachment removal using different tools: OneGloss (B-1, B-2), Enhance (C-1, C-2), SM 104 (D-1, D-2), Sof-Lex disc (E-1, E-2), Tungsten carbide bur (F-1, F-2), Zirconia bur (G-1, G-2), and White stone bur (H-1, H-2). For each tool, the upper row (×150 magnification) shows a general view of the surface, while the lower row (×2,400 magnification) illustrates detailed surface morphology.

Enamel loss measurement

Measurement of enamel loss depth showed an ICC close to 1.00 (95% CI [0.98–0.99]), indicating a very high level of inter-rater reliability. The depth of enamel loss caused by different adhesive removal tools is displayed in Fig. 4. One-way ANOVA and Tukey’s post hoc analysis revealed a statistically significant difference in enamel loss among the groups (p = 0.000, η2 = 0.724). Group 7 exhibited the highest enamel loss compared to all other groups (0.115 ± 0.054 mm), followed by Group 5 (0.042 ± 0.013 mm). There was no significant difference in enamel loss among Groups 1-4 and 6, with mean values of 0.010 ± 0.005 mm, 0.016 ± 0.007 mm, 0.018 ± 0.010 mm, 0.016 ± 0.006 mm, and 0.027 ± 0.011 mm, respectively.

Figure 4 Depth of enamel loss measurement (mm).

The depth of enamel loss across different groups was analysed using a one-way ANOVA test. Means with the same lowercase letters were not significantly different according to Tukey’s test.

Attachment removal time and risk-benefit analysis

Mean removal times varied significantly among the seven tools tested (p = 0.000, η2 = 0.770). The shortest times were observed in Group 1 (45.9 ± 6.7s), Group 2 (44.7 ± 7.8s), Group 3 (49.6 ± 6.0s), and Group 7 (54.0 ± 6.7s), according to Tukey’s post hoc analysis. Group 4 and Group 5 exhibited intermediate removal times, while Group 6 was significantly slower (84.4 ± 10.8s) (Fig. 5).

Figure 5 Attachment removal time (s).

Attachment removal time across different groups was analysed using a one-way ANOVA test. Means with the same lowercase letters were not significantly different according to Tukey’s test.

To contextualize the findings, a risk–benefit matrix was constructed, with each tool’s scores across the three criteria summarized in Table 3. OneGloss, Enhance, and SM 104 occupied the optimal region of the matrix, demonstrating high performance in all three domains: fast removal, minimal enamel loss, and smooth surface finish. In contrast, the white stone bur, despite being one of the fastest tools, caused significant enamel loss and created the roughest surface (0.115 ± 0.017 mm). Tools such as the Sof-Lex disc and zirconia bur showed excellent enamel preservation but were less time-efficient. Overall, the matrix effectively highlights instruments that provide a clinically favorable balance between efficiency and enamel safety (Fig. 6).

Table 3 Risk-benefit scoring of composite removal tools based on Tukey’s post hoc test results.

Tool	Risk		Benefit	
	Enamel loss score	Surface roughness score		Time efficiency score	
OneGloss	1	1		5	
Enhance	1	1		5	
SM 104	1	1		5	
Sof-Lex disc	1	1		3	
Tungsten carbide bur	3	1		3	
Zirconia bur	1	1		1	
White stone bur	5	5		5	

Figure 6 Three-dimensional risk–benefit matrix for seven composite removal instruments.

The horizontal axis represents time efficiency (higher scores indicate faster removal), the vertical axis represents enamel loss (higher scores indicate greater loss), and bubble size denotes surface roughness (larger bubbles indicate rougher surfaces).

Discussion

In this study, we evaluated the effect of seven composite attachment removal instruments, all used in a one-step procedure with condensable composite, on enamel integrity. To the best of our knowledge, this is the first study to compare composite removal tools specifically for clear aligner attachment removal and to apply a risk–benefit model to identify the most suitable tool for clinical use.

Our results showed that single-step polishing systems containing aluminum oxide fillers, including OneGloss, Enhance, and SM 104, matched the white stone in removal time (under 60 s) while producing less enamel loss. These systems also outperformed the Sof-Lex disc, tungsten carbide bur, and zirconia burs in composite removal speed. Moreover, they did not increase surface roughness postoperatively, as confirmed by our data on surface roughness and SEM image analysis. These findings are consistent with prior research comparing OneGloss, Enhance, and tungsten carbide bur systems (Mohammed et al., 2022), and with the study by Janiszewska-Olszowska et al. (2015) which found that OneGloss better preserved tooth structure compared to tungsten carbide burs. Although no statistically significant difference in surface roughness was detected by profilometer measurements among OneGloss, SM 104, and Enhance, SEM image analysis revealed that OneGloss and SM 104 produced superior surface smoothness, characterized by shallow, parallel grooves and the absence of crater formation. The average operating times for OneGloss, SM 104, and Enhance in our study were less than 50 s, each resulting in enamel loss of less than 20 µm. In contrast, the study by Ryf et al. (2012) which employed a two-step adhesive removal system following bracket debonding, reported an average procedure time exceeding 80 s, with a mean enamel loss depth of approximately 50 µm. Similarly, Fan, Chen & Huang (2017) reported that using OneGloss as a single-step polishing system for resin removal after bracket debonding required more than 60 s. These differences suggest that the removal of resin composite used in clear aligner therapy may be less time-consuming and cause less enamel damage compared to bracket-bonding resin. This may be attributed to the fact that bracket bonding resins typically require more extensive light curing from multiple angles to ensure complete polymerization, making them harder to remove (Ghaleb et al., 2024; Ocak, Gorucu-Coskuner & Aksu, 2025). Moreover, the mechanical debonding itself may contribute to iatrogenic enamel damage due to the force applied during bracket removal (Dumbryte et al., 2015).

White stone bur, used with a high-speed handpiece, demonstrated superior time efficiency compared to the Sof-Lex polishing disc, tungsten carbide bur, and zirconia bur. This aligns with the study by Thawaba et al. (2023), which found the white stone bur had the fastest adhesive removal time post-bracket debonding, followed by zirconia and tungsten carbide burs. However, in our study, zirconia burs required more time than tungsten carbide burs, likely due to differences in bur morphology between the tools used in this study and those used in the previous study, which significantly affects cutting efficiency (Di Cristofaro, Giner & Mayoral, 2013). Despite its time-saving advantage, the White stone bur caused greater surface roughness and enamel loss than all other systems tested. SEM image analysis also revealed significant enamel damage, characterized by prominent craters and deep scratches. Given that the outer enamel is the most mineralized layer and plays a critical role in protecting against caries and sensitivity (He et al., 2011; Li et al., 1994), such damage is clinically significant. Additionally, increased surface roughness promotes plaque accumulation (Mei et al., 2009; Tang et al., 2009), with studies indicating that bacterial retention becomes pronounced when Ra exceeds 0.2 µm (An et al., 1995; Bollen et al., 1996; Wang et al., 2015). Therefore, despite its speed, the White stone bur is not recommended for attachment removal in clear aligner therapy.

It is worth noting that previous studies have reported conflicting results regarding surface roughness. For instance, Shah et al. (2019) used a multi-step Sof-Lex protocol and concluded that OneGloss and Enhance produced smoother surfaces. Conversely, a study by Almudhi et al. (2023) reported that Sof-Lex discs produced smoother surfaces than both Enhance and OneGloss polishings. In our study, Sof-Lex was used as a single-step system (coarse disc only), and no significant difference in surface roughness was found between it and the OneGloss or Enhance systems quantitatively and qualitatively. These discrepancies highlight the substantial influence of operator-related factors on polishing outcomes. Surface roughness results can be highly sensitive to how a tool is applied, including the amount of pressure exerted, polishing duration, angulation of the tool against the tooth surface, and even the operator’s clinical experience or skill level. Variations in these factors across studies may partly explain the inconsistent findings despite the use of similar materials or protocols. To minimize this source of bias, all polishing procedures in our study were performed by a single trained investigator under standardized conditions. Nonetheless, the potential for operator variability remains an important limitation and underscores the need for caution when comparing surface roughness outcomes across different studies.

While removal time is often prioritised in clinical settings, our findings emphasise the importance of balancing speed with enamel preservation. To assist in this balancing act, we introduced a 3D risk–benefit matrix that visually synthesizes data on time efficiency, enamel loss, and surface quality for each tool. This matrix revealed that although the white stone bur demonstrated high time efficiency, it also caused the most severe enamel damage, making it a high-risk choice despite its procedural speed. On the other hand, OneGloss and SM 104 not only provided rapid removal but also excelled in preserving enamel morphology and surface smoothness, which are critical for long-term dental health and esthetics. This visual format offered a nuanced view of each tool’s clinical performance, allowing clinicians to visualize trade-offs and select the most balanced option for attachment removal. In daily orthodontic practice, clinicians can use this framework to quickly identify the most appropriate attachment removal tool based on patient-specific priorities. For example, in patients with thin enamel or high esthetic expectations, instruments located in the matrix quadrant favoring minimal enamel loss and smoother surfaces (OneGloss, SM 104) would be preferred. In contrast, when chairside efficiency is a higher priority (e.g., in pediatric or time-sensitive settings), tools that reduce procedure time, even at a slight trade-off in surface quality, may be considered. This structured framework supports individualized, evidence-based decision-making and can also be used as an educational tool for training clinicians to weigh clinical trade-offs systematically.

Compared to previous studies on post-bonding adhesive removal, a major strength of our study lies in the use of uniformly sized attachments across all specimens. This standardization allowed for a more objective and consistent comparison of composite resin removal tools, in contrast to studies that rely on the qualitative Adhesive Remnant Index (ARI) for evaluation (Fan, Chen & Huang, 2017; Ghaleb et al., 2024; Ryf et al., 2012). Furthermore, the incorporation of the oral scanner and advanced 3D measurement technology enabled the precise quantification of subtle enamel surface changes that are often undetectable in clinical settings. This approach also minimized reliance on cumbersome equipment typically required for traditional impression and casting methods (Ryf et al., 2012).

However, our study has several limitations. First, we did not evaluate the influence of composite type on removal efficiency and its impact on enamel integrity. The type of composite resin may significantly affect removal difficulty. For instance, packable composites, which contain a high concentration of filler particles, are known for their increased viscosity and durability (Brandao et al., 2005). These properties may necessitate higher compressive forces during removal, increasing surface contact and potentially contributing to greater enamel roughness after debonding. Further studies are needed to confirm this hypothesis. Additionally, our research did not assess the effect of aging in the oral environment on composite materials or the removal process. Environmental factors such as temperature fluctuations, humidity, and microbial activity can contribute to the degradation of composite resins over time. These changes can alter the resin’s mechanical properties (Drummond, 2008; Bationo et al., 2020) and bond strength to enamel (Oesterle & Shellhart, 2008), potentially making removal more difficult and increasing the risk of enamel damage. As a result, the findings of this study may underestimate the complexity and extent of enamel loss associated with attachment removal in real clinical conditions. To better understand the actual impact of environmental factors and aging on the risk of enamel damage during attachment removal, further research is warranted. Another critical consideration during attachment removal is the generation of airborne particulate matter. As highlighted by Eliades & Koletsi (2020), minimizing the release of airborne nanoparticles during the debonding process is essential to reduce potential respiratory risks for both dental professionals and patients. This issue is particularly relevant in aligner treatments, where bulky composite attachments, which have significantly greater surface area and volume than the adhesive remnants from bracket removal, can generate more airborne particles during grinding (Turkoglu & Atik, 2025). Finally, one methodological limitation is our use of a combination of SEM and profilometry for quantitative and qualitative assessment of enamel surface changes. While SEM provides high-resolution imaging, it is inherently two-dimensional and cannot fully capture variations in surface depth or height. Additionally, only two representative samples from each group were selected for SEM analysis, which limits the generalizability of the morphological observations and necessitates cautious interpretation of these qualitative findings. Other methods, such as atomic force microscopy (AFM), 3D optical profilometer, and confocal laser microscopy, by contrast, offer nanometer-scale resolution, enable 3D surface profiling, and require minimal sample preparation (Karan, Kircelli & Tasdelen, 2010; Turkoglu & Atik, 2025). Future studies may, therefore, benefit from integrating these 3D technologies to enhance surface characterization precision.

Conclusion

Our study demonstrated that, among the seven one-step adhesive removal tools evaluated, OneGloss and SM 104 caused minimal enamel damage and restored the enamel surface to a roughness level comparable to the pre-attachment condition, while also being the most time-efficient. The application of a risk–benefit model offers clinicians a practical framework for selecting the most appropriate tool by balancing efficiency, enamel preservation, and surface smoothness. Clinically, this approach can support individualized, evidence-based decision-making, especially in patients with high esthetic demands or enamel vulnerabilities. Given the multifactorial nature of composite attachment removal, including variables such as composite type, tooth morphology, and intraoral aging, future studies are warranted to validate these findings under different clinical conditions and across broader patient populations.

Supplemental Information

Supplemental Information 1 Attachment removal time, enamel loss, and surface roughness of the investigated teeth

We sincerely thank Ms. Phung Thi Nga and Ms. Vuong Thi Quynh Trang for their valuable technical support during data collection. We also acknowledge the use of artificial intelligence (ChatGPT-4o) for language editing assistance during the preparation of this manuscript.

Additional Information and Declarations

Competing Interests

Author Contributions

Human Ethics

Data Availability

The authors declare there are no competing interests.

Anh Viet Nguyen conceived and designed the experiments, performed the experiments, analyzed the data, prepared figures and/or tables, authored or reviewed drafts of the article, and approved the final draft.

Trang Thi Nguyen conceived and designed the experiments, performed the experiments, analyzed the data, prepared figures and/or tables, authored or reviewed drafts of the article, and approved the final draft.

The following information was supplied relating to ethical approvals (i.e., approving body and any reference numbers):

Ethical approval for this study was granted by Phenikaa University (pu2024-3-d-02).

The following information was supplied regarding data availability:

The raw data is available in the Supplemental File.

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
