# Peer review of "Evaluation of enamel integrity after clear aligner attachments removal: a risk–benefit comparison of one-step tools"

_PeerJ, doi:10.7717/peerj.20200_

## Round 0.1 · original submission · Minor Revisions

Reviewer 1 ·

Basic reporting

The manuscript is written in proper and fluent English. There are only some minor typos and errors. The references are up-to-date, and sufficient background information is provided. The article adheres to proper writing conventions, and the results are presented in alignment with the stated hypothesis.

Experimental design

The experimental design is well described and has a meaningful structure. Methodological explanation is sufficient. Only 2 samples from each group were included in the SEM analysis, which requires a careful approach to the results.

Validity of the findings

When examining enamel damage, the effect of aging along with the duration of the attachment on the tooth limits the results. This issue is partially discussed in the manuscript. The inadequacy of the number of samples included in the SEM evaluation should also be presented as a limitation. It would be useful to mention that operator effect in debonding protocol is also involved.

Additional comments

I congratulate you for your diligent work. I recommend correcting the English spelling mistakes.

Reviewer 2 ·

Basic reporting

Validity of the Work
Strengths:
• The experimental design is solid, with proper controls and statistical analysis.
• A robust sample size calculation was provided.
• Use of profilometry, SEM, and 3D scan analysis enhances methodological reliability.

Limitations (to be addressed):
• The methodology would benefit from clarification on operator calibration and intra/inter-examiner reliability for enamel loss measurements.
• Lack of blinding in measurement steps (e.g., surface roughness or time recording) could introduce bias.
• While the inclusion of seven tools is comprehensive, the absence of a true control (e.g., untreated enamel ) could affect comparative interpretation.

Experimental design

Originality and Significance
Strengths:
• This is the first study to compare various one-step composite removal tools specifically for clear aligner attachments.
• The risk–benefit matrix is a practical innovation with strong clinical utility.

Suggestions:
• The discussion should more directly elaborate how this matrix could be integrated into daily orthodontic decision-making.
• Please emphasize how your findings differ from previous bracket debonding studies.

Validity of the findings

Clarity and Presentation
Strengths:
• The manuscript is well-structured and generally well-written.
• Terminology is appropriate for the scientific audience.

Improvements:
• A few minor grammatical/wording issues need revision (e.g., “the Group 2 exhibited.” should be “Group 2 exhibited”).
• Ensure uniform reference formatting (e.g., “Keet al. et al.” → likely meant “Ke et al.”).


Data Quality and Interpretation
Strengths:
• The combination of quantitative (Ra, enamel loss, time) and qualitative (SEM) methods supports comprehensive evaluation.
• Statistical analysis is thorough and appropriate.

Recommendations:
• Consider reporting effect sizes in addition to p-values for better interpretation of clinical significance.
• Some figures (e.g., SEM images) may benefit from more detailed captions or insets showing magnification.

Additional comments

Ethics and Transparency
Strengths:
• Ethical approval is clearly stated.
• Data collection appears compliant with research standards.
Suggestions:
• Add a data availability statement.


Final Recommendation
This is a clinically relevant and methodologically sound study that deserves publication after minor revisions. The integration of a risk–benefit matrix to guide clinicians is a novel and highly practical addition to the literature.
Please revise minor formatting/language issues and consider adding deeper commentary on clinical applications and measurement reliability.

·

Basic reporting

1.1. English language and clarity need revision.
Although the manuscript is largely comprehensible, the academic English requires polishing. Several phrases contain grammatical inconsistencies or awkward structure that may hinder reader comprehension.
• Example: "The Group 2 exhibited a surface characterised..." should be "Group 2 exhibited a surface characterized..."
• Suggestion: Have the manuscript reviewed by a native English speaker or professional editing service.
1.2. References contain redundant or incorrect formatting.
Several citations are rendered incorrectly, such as Keet al. et al. or Gazzaniet al. et al.. This appears to be a systematic error with the citation tool used.
• Suggestion: Carefully revise all in-text references to remove duplications and formatting issues.
1.3. Minor structural redundancies.
Some terms and expressions are unnecessarily repeated (e.g., the phrase “risk–benefit model” appears in almost every section without added clarity). This should be streamlined.

Experimental design

2.1. Well-designed and relevant study.
The methodology is rigorous, and the use of a standardized attachment design and consistent removal protocol enhances reproducibility.

Strength: Use of 3D superimposition and SEM improves measurement objectivity.
2.2. Ethical approval and informed consent are clearly stated.
This meets the journal's requirements for studies involving human-derived material.

2.3. Sample size justification is appropriate.
The use of G*Power for determining group sizes is commendable and shows statistical foresight.

Validity of the findings

3.1. Results are well supported by the data.
There is clear alignment between the experimental observations (Ra values, SEM images, enamel loss measurements) and the conclusions drawn.
3.2. The risk–benefit matrix is a valuable clinical tool.
The visual integration of risk and efficiency into a 3D matrix adds real-world applicability to the findings.
3.3. Limitations are well acknowledged.
The discussion section adequately addresses limitations such as the lack of evaluation of composite type or real-time intraoral degradation effects.

Additional comments

Strengths of the manuscript:
• The study addresses a relevant clinical issue with practical implications for orthodontists.
• The standardized method for attachment placement and removal improves consistency.
• The inclusion of multiple assessment modalities (profilometry, SEM, 3D deviation analysis) strengthens the findings.
Suggestions for improvement:
1. Language editing is essential to enhance readability and meet PeerJ’s standard for clear academic writing.
2. Correct corrupted characters throughout the manuscript.
3. Fix reference formatting, especially to remove “et al. et al.” redundancies.
4. Clarify certain transitions in the introduction and discussion. For example, the leap from enamel damage to clinical decision-making tools could be more smoothly developed.
5. Consider expanding the conclusion, to briefly reiterate the clinical implications and call for future studies.

---

## Round 0.2 · accepted · Accept

All reviewers' concerns have been adequately addressed during the revision and the manuscript can be accepted for publication in its current form.

Reviewer 1 ·

Basic reporting

All suggested corrections have been carefully implemented. A current and important topic has been studied. I congratulate them on this work and appreciate their efforts in the revision. Their work in all areas is sufficient.

Experimental design

Necessary revisions have been made.

Validity of the findings

Data sharing is sufficient and understandable.

Additional comments

I congratulate the authors and wish them success in their future studies.

Reviewer 2 ·

Basic reporting

The manuscript is well-written, clear, and well-structured. The language is professional, and the literature review provides sufficient context. Figures and tables are relevant and well-labeled, though improving the SEM image contrast would enhance clarity. References are up-to-date and appropriate.

Experimental design

The study follows a clear and well-structured experimental design. The research question is well-defined, and the chosen methodology is appropriate to address the study objectives. Sample selection, data collection, and analysis are adequately described, allowing reproducibility. The statistical methods are sound and properly applied. Overall, the experimental design is solid and supports the reliability of the findings.

Validity of the findings

The findings are well-supported by the data and align with the study objectives. Statistical analysis is appropriate and strengthens the conclusions. Results are clearly presented, and the discussion effectively relates them to existing literature. No overinterpretation is observed, and the conclusions are consistent with the evidence provided. Overall, the findings are valid and scientifically sound.

Additional comments

The manuscript is well-prepared, scientifically sound, and addresses an important topic in dental research. The methodology is appropriate, the results are clearly presented, and the discussion is relevant and well-supported by the data.

·

Basic reporting

The manuscript is written in clear and professional academic English. The grammatical and stylistic issues noted by previous reviewers have been corrected. References are current, relevant, and well integrated throughout the discussion. The structure of the article adheres to academic standards, and figures and tables are properly presented.

The inclusion of the 3D risk–benefit matrix enhances the clarity of the results and their clinical applicability. The manuscript is self-contained, and the results are directly linked to the stated hypothesis.

Minor suggestion: Ensure that figure legends clearly indicate SEM magnification levels (×150 and ×2400), and add a “Data Availability” section at the end of the manuscript.

Experimental design

This is an original study that fits well within the scope of the journal. The research question is clear, clinically relevant, and addresses a gap in the literature regarding the removal of attachments in clear aligner therapy.

The experimental design is rigorous, with appropriate sample size calculation, operator calibration, and control measures. The methodological description is sufficiently detailed to allow replication.

Minor suggestion: Add a clarifying note in the acknowledgments section regarding the use of AI tools, specifying that they were used solely for language editing, in accordance with PeerJ’s transparency guidelines.

Validity of the findings

The data are consistent, statistically sound, and clearly presented. The combination of qualitative (SEM) and quantitative (Ra, enamel loss, removal time) analyses strengthens the validity of the findings.

The conclusions are well supported by the data and remain within the scope of the study design. Including effect sizes (partial η²) was appropriate and allows for improved clinical interpretation beyond p-values.

Additional comments

Congratulations to the authors on a methodologically sound and clinically valuable study. The risk–benefit model is a novel and practical contribution to clinical decision-making in orthodontics with clear aligners. The manuscript is nearly ready for publication following the implementation of a few minor formatting revisions.